# Screening of *Bacillus velezensis* E2 and the Inhibitory Effect of Its Antifungal Substances on *Aspergillus flavus*

**DOI:** 10.3390/foods11020140

**Published:** 2022-01-06

**Authors:** Shengjie Li, Xingang Xu, Tianyuan Zhao, Jianing Ma, Luning Zhao, Qi Song, Weihong Sun

**Affiliations:** 1College of Food and Biological Engineering, Jiangsu University, Zhenjiang 212013, China; lsjash@163.com (S.L.); zhaoluning@ujs.edu.cn (L.Z.); 2College of Agricultural Engineering, Jiangsu University, Zhenjiang 212013, China; 17806275211@163.com (X.X.); zhaotianyuan0711@163.com (T.Z.); majianing7581@163.com (J.M.); songqi_ujs@163.com (Q.S.)

**Keywords:** *Aspergilus flavus*, lipopeptides, antifungal activity, aflatoxin biosynthesis

## Abstract

*Aspergilus flavus* is the main pathogenic fungus that causes food mold. Effective control of *A. flavus* contamination is essential to ensure food safety. The lipopeptides (LPs) produced by *Bacillus* strains have been shown to have an obvious antifungal effect on molds. In this study, an antagonist strain of *Bacillus velezensis* with obvious antifungal activity against *A. flavus* was isolated from the surface of healthy rice. Using HPLC-MS analysis, the main components of LPs produced by strain E2 were identified as fengycin and iturins. Further investigations showed that LPs could inhibit the spore germination, and even cause abnormal expansion of hyphae and cell rupture. Transcriptomic analyses showed that some genes, involved in ribosome biogenesis in eukaryotes (NOG1, KRE33) and aflatoxin biosynthesis (aflK, aflR, veA, omtA) pathways in *A. flavus* were significantly down-regulated by LPs. In conclusion, this study provides novel insights into the cellular and molecular antifungal mechanisms of LPs against grain *A. flavus* contamination.

## 1. Introduction

*Aspergilus flavus* is a common saprophytic fungus, which can colonize a range of economically important crops [1]. In addition to causing great economic losses, the contamination of food and feed by *A. flavus* also causes public health concerns because this fungi can produce mycotoxins (aflatoxin) [2,3]. Aflatoxins have been shown to possess carcinogenic, teratogenic, and mutagenic effects that are harmful to the human body [4]. Therefore, an efficient mechanism for controlling *A. flavus* contamination is urgently needed. Currently, the common method for controlling *A. flavus* contamination is using synthetic chemical fungicides. However, this method is limited because it can lead to the presence of fungicide residues in food and fungal resistance, as well as posing a threat to people’s health [5,6]. In contrast, biological control provides a safe and eco-friendly method for inhibiting the growth of fungal pathogens [7].

To date, many microbial strains, such as yeast, bacteria, and filamentous fungi, have been used for the control of mold contamination. The biological control mechanism mainly involves the production of direct antifungal substances, such as non-volatile metabolites [8], volatile organic compounds (VOCs) [9,10] and extracellular lytic enzymes [11]. Among the microbial strains associated with the biological control mechanism, the broad-spectrum antagonistic *Bacillus* has gradually become one of the most valuable, because of its good biological activity, high inhibitory activity to a variety of pathogenic microorganisms, and convenient mass production [12,13]. Other strains that have also been reported to have good antifungal activity include *Bacillus subtilis* fmbJ [14], *Paenibacillus polymyxa* APEC136 [12], *Bacillus atrophaeus* strain B44 [15], and *Bacillus amyloliquefaciens* MG3 [16].

Antifungal LPs are important metabolites produced by some strains of the genus *Bacillus* and they play an important role in disease suppression [17]. Recent studies have shown that LPs can be produced by various *Bacillus* strains, such as *Bacillus*
*subtilis* [18], *Bacillus*
*amyloliquefaciens* [19], *Bacillus*
*thuringiensis* [20], and *Bacillus*
*cereus* [21]. The antimicrobial LPs mainly include surfactin, fengycin, and iturin [22].

So far, molecular mechanisms associated with the antifungal activity of LPs against *A. flavus* are still to be elucidated. This study aimed to isolate and identify antagonistic bacterial against *A. flavus*, characterize the substance responsible for its antifungal activity, and explore the possible mode of action of the identified antifungal substance against pathogenic fungi.

## 2. Materials and Methods

### 2.1. Fungal Pathogen

*A. flavus* was isolated from the surface of moldy rice and identified by morphological characteristics and DNA sequencing. It was streaked on the surface of potato dextrose agar (PDA) plates, incubated at 28 °C for 5 days, and temporarily stored at 4 °C for later use [23]. A small quantity of sterile distilled water (SDW) was added to the plate to wash off the spores. The mold suspension was filtered with three layers of lens cleaning paper to remove the hyphae and a hemocytometer was used to adjust it to the desired concentration [10].

### 2.2. Isolation and Identification of Antagonistic Bacterial Strains

The isolation source of antagonistic bacteria was healthy rice, paddy leaves, and rhizosphere soil collected from the rice fields in the city of Zhenjiang City, China. Isolation and purification was carried out according to the method of Yang [8]. The isolated and purified strains were inoculated in LB medium at 28 °C in a rotatory shaker (180 rpm), for 24 h. Samples were collected and stored at 4 °C as for later use. The assessment of the inhibitory effects of the isolated bacterial strains was performed through cup–disc methods [24]. Holes with a diameter of 6 mm were punched in the center of the PDA plate (90 mm) with 4 holes equally spaced about 20 mm away from the center hole, and the agar in the holes was removed. 40 μL 1 × 10^5^ CFU/mL of *A. flavus* spore suspension and 40 μL 1 × 10^8^ CFU/mL of antagonistic bacterial suspension were selected for the test. These were incubated at 28 °C for 4 days. It was then observed and the size of the mycelial diameter of the pathogenic bacteria was measured. Three replicates were analyzed per treatment.

The bacteria with the best antagonistic effect were analyzed as follows. The morphological traits of strain E2 were observed and recorded after incubation on LB agar at 28 °C for 24 h. Physiological and biochemical tests of strain E2 for bacterial identification were performed as described in Bergey’s Manual of Determinative Bacteriology [25].

Total DNA was extracted according to the methods of Linlin [26]. Universal primers 27F (5′-AGAGTTTGATCCTGGCTCAG-3′) and 1492R (5′ACGGCTACTGTACGACT- -T-3′) were used to amplify the 16S rDNA gene. PCR products were sequenced by Sangon Biotech Co. Ltd. The results of sequencing were compared using NCBI. Related sequences were downloaded and used to construct a phylogenetic tree with the Mega 6.0 software [27].

### 2.3. Extraction of Antifungal Substances

A single bacterial colony was transferred into LB medium and incubated in a rotatory shaker (180 rpm) for 24 h, as seed liquid. The seed liquid (1%) was transferred to the LB medium and incubated at 28 °C in a rotatory shaker (180 rpm) for 72 h. The fermentation broth was centrifuged at 10,000 rpm for 15 min at 4 °C. The pH of the collected supernatant was adjusted to 2.0 using 6 moL/L concentrated HCl, and it was placed at 4 °C for 24 h. The samples were centrifuged again at 10,000 rpm for 15 min at 4 °C, after 24 h [18]. The crude LPs were collected and then extracted by methanol for 6–8 h, and filtered through syringe-type 0.22 μm microbial filters. A vacuum concentrator was used to remove methanol from the mixture to obtain the yellowish-brown substance [28]. The extracted substance was dissolved in SDW and adjusted to neutral with 1 moL/L NaOH. After pre-freezing, it was freeze-dried in a vacuum freeze dryer for 48 h. The light brown loose powder solid was LPs.

### 2.4. LPs Minimal Inhibitory Concentration

A defined concentration (100 μL) of *A. flavus* spore suspension (1 × 10^5^ CFU/mL) was spread on PDA plates (90 mm), and 5 holes (6 mm) were punched following the method described in Section 2.2. Different concentrations 0.25, 0.5, 1, 2.5, 5, 10, and 15 mg/mL of LPs and a 10^9^ CFU/mL bacterial suspension were into each hole with a respective volume of 40 μL. An equivalent amount of SDW was added in place of the LPs suspension in one of the holes and was used as a non-treated control. The inhibitory diameter of hyphae was measured under different LPs treatment after 5 days. Three replicates were analyzed per treatment.

### 2.5. Effect of LPs on A. flavus Spore Germination

LPs were added into erlenmeyer flasks containing 20 mL of PDB medium to make the final concentration to 12.5, 25, 50, 100, 200, and 400 µg/mL. 2 mL of 1 × 10^7^ *A. flavus* spore suspension was inoculated into each, and the PDB without LPs was used as non-treated control. After incubating at 28 °C and 75 rpm, for 10 h, the spore germination rate of *A. flavus* was observed and calculated. When the length of the germ tubes was equal to the maximum size of the swollen spore, the spores were considered to have germinated [29]. At least 100 spores were observed each time using a light microscope. The spore germination rates were calculated as follows: spore germination rate (%) = (number of germinated spores/total number of observed spores) × 100% [16]. Three replicates were analyzed per treatment.

### 2.6. Effect of LPs on A. flavus Mycelium Growth

LPs concentrations of 0, 12.5, 25, 50, 100, 200, 500, and 1000 µg/mL were prepared in PDA medium and poured into plates. A single hole was punched at the center of the plates with a sterile punch (6 mm), and the agar in the hole was removed. 40 µL 1 × 10^6^ CFU/mL of *A. flavus* spore suspension was added to each hole. The PDA medium without LPs was used as the control. After incubating at 28 °C for 5 days, the diameter of the mycelium was observed and calculated. Three replicates were analyzed per treatment.

### 2.7. Microscope and Scanning Electron Microscopy (SEM)

A total of 1 × 10^6^ CFU/mL *A. flavus* spore suspension was added to the PDB medium amended with an LPs concentration of 500 µg/mL. The PDB medium without LPs was used as the control. After incubating at 28 °C for 38 h, samples were prepared according to Ye [30]. The collected *A. flavus* hyphae were fixed in glutaraldehyde solution at 4 °C overnight. The specimens were washed 3 times with 0.1% moL/L PBS for 15 min each time. Then, 30, 50, 70, 90, and 95% ethanol were used to dehydrate the specimens in gradients for 15 min each. The specimens were dehydrated with 100% ethanol three times for 20 min each time. Next, the alcohol was replaced with pure tertiary alcohol three times, and the standing time for each session was 15 min. Finally, the mixed mycelial pellet and tert-butanol suspension were sucked and dropped on the sample table covered with a cover glass. They were vacuum dried in a freeze dryer that was pre-cooled for 1 h. Specimens were taken out after the air pressure dropped below 10 Pa. The dehydrated specimens were coated with gold-palladium and observed under the thermal field emission scanning electron microscope (JSM-7001F).

### 2.8. Analysis of LPs by HPLC-MS

LPs were qualitatively analyzed by a liquid Chromatograph Mass Spectrometer (*thermo LXQ LC/MS). The HPLC conditions were as follows: C18 column, 5 μm, 4.6 mm × 250 mm; mobile phase A [distilled water containing 0.08% (*v/v*) formic acid]; mobile phase B [acetonitrile containing 0.08% (*v/v*) formic acid]; and flow speed, 0.6 mL/min; Linear gradient programmes were set from 40% of phase B to 60% of phase B during a 35 min period; 30 °C and UV detection at 210 nm. Mass spectrometry conditions: ion source, ESI; ion spray voltage, 3500 V; temperature, 325 °C; mass spectra, 100–1500 *m/z*. Data acquisition and processing were performed using Xcalibur 4.2.

### 2.9. Transcriptomics Study on the Effect of LPs on A. flavus

#### 2.9.1. RNA Extraction and Transcriptome Sequencing

According to the inoculation amount of 2%, 1 × 10^6^ CFU/mL *A. flavus* spore suspension was inoculated into PDB medium with an LPs concentration of 1 mg/mL. The PDB medium without LPs was used as the control. *A. flavus* spores were incubated at 28 °C, 180 rpm, and protected from light for 38 h. Mycelium was collected and washed three times with SDW and frozen with liquid nitrogen. A total RNA of the sample was extracted according to the instructions of Trizol reagent kit (Invitrogen, Carlsbad, CA, USA). The concentration, purity, and integrity of RNA in the sample were detected with Agilent 2100, and the qualified samples were sent to Gene Denovo Biotechnology Co. (Guangzhou, China) for the construction and sequencing of the cDNA libraries. Ensembl_release51 was selected as the reference genome.

#### 2.9.2. Transcriptome Sequencing

In order to ensure data quality, fastp was used to analyze the original raw reads. Low-quality data were filtered out and clean reads were obtained. DESeq2 software was used to screen differentially expressed genes. Genetic parameter of false discovery rate (FDR) ≤ 0.05 and absolute fold change (FC) ≥ 4 were considered to be differentially expressed gene [31]. The expressed genes (FDR ≤ 0.05 and absolute FC ≥ 4) were functionally annotated according to three databases, Kyoto Encyclopedia of Genes and Genomes (KEGG) and Gene Ontology (GO) [32]. Moreover, both upward and downward gene expression profiles were subjected to GO and KEGG enrichment analysis. In addition, GO and KEGG enrichment analyses were performed on both the upward and downward gene expression profiles.

#### 2.9.3. RT-qPCR

Genes with significant differences were selected from the resulting transcriptomic structure. Primer design was performed by the method of Xiong [33]. The actin gene was used as a reference gene for relative gene expression analysis. The primers used were listed in Table 1. Referring to the method of Redshaw [34] and using a 25 μL reaction system, the parameters in the real-time fluorescence quantitative PCR analyzer were set as follows: pre-denaturation 30 s at 95 °C, denaturation 5 s at 95 °C, then annealing 30 s at 62 °C, and finally elongation 20 at 72 °C, 40 cycles; The dissolution curve is: 95 °C, 15 s; 60 °C,1 min; 95 °C, 15 s.

### 2.10. Statistical Analyses

All test data were statistically analyzed by Excel 2010 and SPSS 22.0 software. Duncan’s Multiple Range Test was used to analyze the significance of the difference, and the level of significance was set to *p* < 0.05.

## 3. Results

### 3.1. Screening and Identification of Antagonistic Strains against A. flavus

A total of 84 strains of microorganisms were isolated from the rice surface, soil, and paddy leaf surface. Through the cup–disc tests, strain E2 with the highest antifungal rate against *A. flavus* was selected as the antagonistic bacteria (Figure 1A). Cells of strain E2 were Gram-stain-positive and had a short rod shape (Figure 1B). The results of some physiological and biochemical tests are shown in Table 2. Except for one indicator of methyl red test which was negative, the results of the other 13 indicators, including the V-P test and the citrate utilization, were all positive. Under the guidance of these tests, the E2 strain was preliminarily speculated to belong to the *Bacillus* spp. [35].

The Blast software on the NCBI website was used to analyze the obtained E2 16S rDNA gene sequence. The E2 strain had more than 99% homology with some *Bacillus* strains in the NCBI database. The phylogenetic tree was constructed as shown in Figure 2. Based on morphological characteristics and molecular biological identification, the antagonistic E2 strain was identified as *Bacillus velezensis*.

### 3.2. The Minimal Inhibitory Concentration of LPs

The inhibitory effect of LPs on *A. flavus* was obviously dependent on its concentration. With the increase in LPs concentration, the inhibitory effect became stronger (Figure 3). When the LPs concentration was 0.5 mg/mL, the inhibition zone produced by its inhibitory effect on *A. flavus* was 8.42 mm, which was the minimal inhibitory concentration of LPs. The diameter of the inhibition zone produced by the LPs concentration reaching 2.5 mg/mL was already higher than the diameter of inhibition produced by the 1 × 10^9^ CFU/mL strain E2 suspension. This showed that the antifungal effect of high-concentration LPs was better than that of bacteria. There was no significant difference in the inhibitory effect of LPs at 10 mg/mL and 15 mg/mL (*p* < 0.05), so the maximum concentration of LPs was 15 mg/mL.

### 3.3. The Inhibitory Effect of LPs on the Germination of A. flavus Spores

When the concentration of LPs was 500 µg/mL, the spore germination rate of *A. flavus* was only 1.33%, and there was almost no germination after incubation for 10 h (Table 3). In the meantime, the spore germination rate of the control group was as high as 75.34%. As the concentration of LPs increased, the germination rate of spores was severely inhibited. This indicated that LPs at an appropriate concentration can effectively inhibit the germination of *A. flavus* spores.

### 3.4. Inhibitory Effect of LPs on Mycelial Growth

After culturing at 28 °C for 5 days, the diameter of the hyphae of *A. flavus* treated with LPs was much smaller than that of the control group (Figure 4). LPs could effectively inhibit the growth of *A. flavus* hyphae and had a very significant inhibitory effect on *A. flavus*. The hyphae of the pathogenic bacteria in the treatment group were thinner, and the hyphae at the edges were more chaotic. When the LPs concentration increased from 0 to 200 µg/mL, the mycelial diameter of *A. flavus* dropped sharply. The mycelial diameter did not decrease until the LPs concentration reached 1000 µg/mL, but a small number of hyphae could still be observed in the hole. This indicated that LPs could inhibit the mycelial growth of *A. flavus*, but could not completely kill it.

### 3.5. Microscope and SEM Evaluation

As shown in Figure 5 and Figure 6, the morphological changes in *A. flavus* hyphae were analyzed in detail with the use of an optical microscope and SEM. The pictures clearly displayed the inhibitory effect of LPs on the growth of *A. flavus*. In the control group, the hyphae of *A. flavus* were dense and produced conidia (Figure 5A). The morphology of the mycelium was regular and full, the structure was complete, and its surface had spikes (Figure 6A–D). In contrast, the hyphae of *A. flavus* treated with LPs were sparse and could not form conidia (Figure 5B). The hyphae were abnormally swollen, twisted, and collapsed. The top of the hyphae was damaged and their surface was smooth (Figure 6E–H).

### 3.6. HPLC-MS Spectrometry Analysis

The results of mass spectrograms detected by LC-MS are shown in Figure 7. It could be seen that the relative molecular masses corresponding to the retention time of 13.25, 15.99, 16.52, 20.95, and 21.85 min were 1043.87, 1057.90, 1057.90 1071.91, and 1071.92, respectively (Figure 7). The mass spectrum peaks corresponded to the ion addition peaks of C14 Iturin A, C15 Iturin A, C15 Iturin A, C16 Iturin A, and C16 Iturin A (Table 4). The relative molecular masses corresponding to the retention time of 24.89, 27.20, and 30.61 min were 1464.20, 1478.20, and 1492.22, respectively (Figure 7). The mass spectrum peaks of them corresponded to the ion addition peaks of C16 Fengycin A, C16 Fengycin C, and C16 Fengycin D (Table 4). The results were similar to those of Youyou Wang [36] and Bo Zhang [37]. These results indicated that strain E2 could produce two LPs; Iturin and Fengycin.

### 3.7. LPs Affected the Transcriptome of A. flavus

#### 3.7.1. Analysis of Differentially Expression Genes

The results showed that under the biological control of LPs, numerous *A. flavus* genes were differentially expressed. The differentially expressed genes between the control group and the treatment group are shown in Figure 8. Each dot in the figure represents a gene. A total of 1500 differentially expressed genes were screened under the criteria of |log_2_ (Fold Change)| ≥ 2 and FDR < 0.05. There were 400 up-regulated expressions and 1100 down-regulated expressions.

#### 3.7.2. GO Enrichment Analysis of Differentially Expressed Genes

All genes and differential genes were classified by GO enrichment and classified into biological process, cell component, and molecular function. The most diverse genes in the biological process were found in the metabolic process (1174) and cellular process (777). Membrane (644) and membrane part (621) were the most diverse genes in the cell component. The most diverse genes in molecular function were catalytic activity (1175) and binding (982) (Figure 9). In most pathways, the number of down-regulated genes was higher than that of up-regulated genes. In contrast, LPs had a stronger inhibitory effect on the gene expression of *A. flavus*.

#### 3.7.3. KEGG Enrichment Analysis of Differentially Expressed Genes

The KEGG enrichment classification results of the screened differential genes showed that a total of 212 differential genes were involved in 45 metabolic pathways (Figure 10). The pathways involving many of up-regulated genes mainly included biosynthesis of amino acids and oxidative phosphorylation. It was speculated that the active ingredients of LPs caused protein damage and the membrane structure was destroyed, thereby the protein synthesis of *A. flavus* was compensatively promoted to meet its growth needs. The pathways involving many of down-regulation of differential genes were mainly concentrated in ribosome biogenesis in eukaryotes and Aflatoxin biosynthesis.

Ribosomes are the cellular factories responsible for making proteins. In eukaryotes, ribosome biogenesis involves the production and correct assembly of four rRNAs and about 80 ribosomal proteins. Even under optimal growth conditions, the lack of these proteins will cause the biosynthesis of ribosomes to stall, and cell growth will stop. According to the results of transcriptome sequencing, the representative genes mpp10, SPAC57A7.06, KRE33, utp10, SPBC4F6.14, NOG2, and NOG1 related to ribosome synthesis in eukaryotes, were all down-regulated (Table 5), and the multiples of difference were exceeded twice.

In order to explore the molecular mechanism of LPs on the synthesis of aflatoxin, the genes related to aflatoxin biosynthesis were screened based on the results of transcriptome sequencing. This demonstrated that the transcription level of most genes, whether it was regulatory genes or structural genes, was down-regulated to varying degrees (Table 6). Among them, aflQ, aflK, ordA, omtA, omtB, aflG, aflN, aflM, aflJ, aflD and aflT were significantly down-regulated.

#### 3.7.4. RT-qPCR Verification of Differentially Expressed Genes

The 8 key genes were screened out and their expression levels were verified by RT-qPCR (Figure 11) KRE33 and NOG1 related to Ribosome biogenesis in eukaryotes were all down-regulated, and the aflR, aflK, omtA, and omtB involved in Aflatoxin biosynthesis were all down-regulated. *A. flavus* growth and secondary metabolism of global regulation of genes veA and laeA were also down-regulated. The results showed that the expression levels of those genes detected by the RT-qPCR were consistent with the transcriptome results. Therefore, the transcriptome results were valid and reliable.

## 4. Discussion

As mentioned above, *A. flavus* can infect rice, peanuts, and corn and produce a mass of conidia [38]. Early prevention of *A. flavus* contamination is of great significance for global food security. Biological control is an effective means to reduce the use of chemicals. Biological control using microbes that were isolated from the environment was an effective and non-toxic approach for controlling disease. Studies have shown that some strains of the genus *Bacillus* have the ability to produce secondary metabolites with strong antifungal activity [39]. Among these antifungal substances, the LPs synthesized through non-ribosomal pathways (Fengycin, Iturin, and Surfactin) have the characteristics of strong antifungal activity, stable properties, and high safety [18,40]. However, their mode of action, especially their molecular mechanism of inhibiting *A. flavus* is not yet clearly understood. In the present study, we screened a batch of bacteria, and among them, strain E2 had the best inhibitory effect on *A. flavus*. The LPs were extracted according to the method of Zhang for follow-up studies [28].

In order to gain some insight into the mechanism of action of LPs on *A. flavus*, we undertook a more specific study on its antifungal effects at the cellular level. It was found that the minimal inhibitory concentration of LPs was 0.5 mg/mL, and its inhibitory effect with the concentration of 2.5 mg/mL was better than that of the bacterial suspension. In addition, the spore germination rate of LPs treatment was much lower than that of the control group. These results demonstrate that one of the mechanisms which enable strain E2 to have an inhibition effect on mold is the production of LPs, which is consistent with previous reports [41]. Combining the results of HPLC-MS and SEM experiments, it is believed that the antifungal effect of LPs depends on Fengycin and Iturin. Iturin has a high hemolytic function and have a strong inhibitory effect on a variety of pathogenic fungi. It mainly causes cell death by promoting the release of macromolecular substances, electrolytes, and the degradation of phospholipids. Fengycins caused the mycelium to be abnormally enlarged and twisted, damaged the cell wall, and formed the cavity [42]. These results indicate that the LPs produced by E2 have a destructive effect on the growth and reproduction of the hyphae of *A. flavus*. Additionally, the hypha of *A. flavus* was damaged when treated with LPs, suggesting that their antifungal activity might be related to the down-regulation of some key protein synthesis genes.

To address this, we analyzed the pathways related to the protein synthesis process. Ribosomal protein was closely related to the growth and development of *A. flavus*. Interestingly, we found that most of the genes related to Ribosome biogenesis in eukaryotes are down-regulated. Among them, KRE33 is essential for the pre-rRNA processing reaction of 18S rRNA synthesis and the assembly of 40S ribosomal subunits. The depletion of KRE33 led to defects in nucleolar assembly, cytokinesis, and cell cycle arrest. Combined with the research of Yanez-Mendizabal [43], it can be speculated that this key gene was down-regulated under the action of the Fengycins, which changed the structure and permeability of cytoplasmic membrane, leading to inhibition of mycelial growth and conidial germination. Nucleolar G protein 1 (NOG1) is a member of the ODN family of GTP-binding proteins, involved in the assembly of the 60S ribosomal subunit precursor [44]. The reduction in its expression led to a dramatic decrease in 60S ribosomal subunits, and over-accumulation of pre-rRNA processing. NOG2 encodes Nog2p. The transient nature of Nog2p and ribosomal precursors may be a GTP-dependent association. Nog2p is an essential protein required for the maintenance of normal rRNA levels [45]. Yan [16] reported that as the concentration of Iturin increased, the protein content of *C. gloeosporioides* gradually decreased. It can be speculated that Iturin inhibited the expression of these genes. The result of the action was the failure of protein synthesis due to the destruction organelle structures. The result of KEGG analysis showed that ribosome biogenesis was the most dysregulated pathway, and suggested that LPs depressed ribosome biogenesis and led to apoptosis of *A. flavus* [46].

Previous studies had confirmed that LPs can effectively inhibit the synthesis of aflatoxin [47], but its molecular mechanism is still unclear. To address this point, the influence of LPs on the pathway synthesizing aflatoxin at the molecular level was further investigated. Some genes involved in the synthesis of AFB1 and other aflatoxins were down-regulated during transcription after treatment with LPs. Among them, veA, as the core protein of the velvet complex (composed of veA, laeA, and velB) [48], regulates the growth and development of *A. flavus*. Additionally, the synthesis of aflatoxins is globally regulated by veA and laeA [49,50]. Additionally, veA is involved in conidiogenesis and the production of sclerotia. AflR and aflS are the biosynthetic regulatory genes of aflatoxin, which play a key role in the regulation of aflatoxin biosynthesis [51,52]. Most genes in the aflatoxin gene cluster are regulated by them [53]. AflK participates in the conversion of VAL to VERB, which is a key step in the formation of aflatoxin. It blocks the bifuran ring of aflatoxin, which is necessary for the binding of aflatoxin to DNA and confers the mode of action of aflatoxin as a mutagen [54]. AflP (omtA) protein is a key enzyme in the late stage of aflatoxin synthesis. These genes are all down-regulated under the action of LPs [55]. LPs reduced the production of aflatoxin by directly inhibiting the process of aflatoxin synthesis. In summary, LPs showed dual inhibitory activity on the growth and reproduction of fungi and the production of aflatoxin. Therefore, LPs produced by *Bacillus velezensis E2* have the potential to prevent contamination by *A. flavus* in the food industry.

## 5. Conclusions

In this study, LPs produced by *Bacillus velezensis E2* were shown to have the potential to control *A. flavus* contamination. The antifungal activity of LPs was associated with the destruction of cell membrane integrity and interference with the normal functioning of ribosomes. Under the treatment with LPs, the expression of genes related to aflatoxin synthesis and its growth was down-regulated, thereby affecting its normal physiological metabolism and the synthesis of secondary metabolites. These results provide a new strategy for preventing *A. flavus* contamination.

## Figures and Tables

**Figure 1 foods-11-00140-f001:**
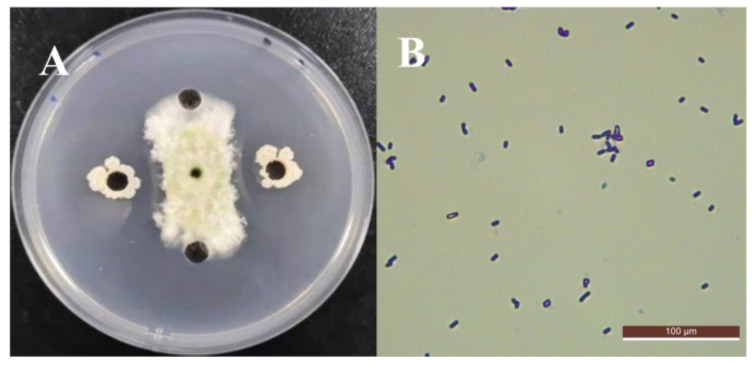
Inhibitory effect of strain E2 on *A. flavus* using cup–disc (**A**), Gram staining results of N2 and its morphology (**B**).

**Figure 2 foods-11-00140-f002:**
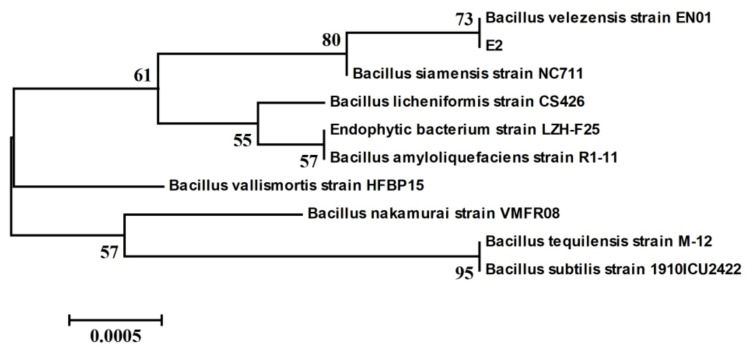
Phylogenetic tree of strain E2 based on 16S rDNA gene sequences. The phylogenetic tree was constructed by the neighbor-joining method using MEGA 6.0 software. The bootstrap values are shown at the branch points.

**Figure 3 foods-11-00140-f003:**
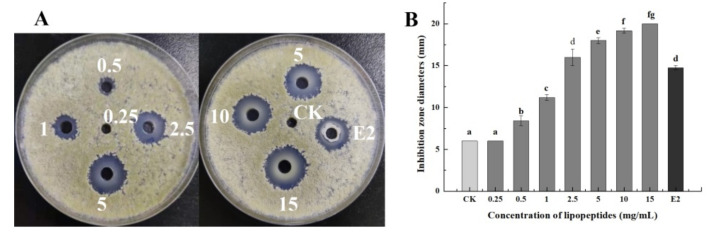
Strain E2 and different concentrations of LPs produced different sized inhibitory zones on the plates containing *A. flavus*. (**A**): Inhibition of *A. flavus* treated with strain E2, LPs (0.25–15 mg/mL), and SDW (CK). (**B**): MIC inhibitory concentrations of LPs. The numbers of 0, 0.25, 0.5, 1, 2.5, 5, 10, 15 mg/mL and CK represent different concentrations of LPs and cell suspensions of strain E2 at the concentrations of 1 × 10^9^ CFU/mL, respectively. Different letters above the bars indicate significant differences (LSD test, *p* < 0.05).

**Figure 4 foods-11-00140-f004:**
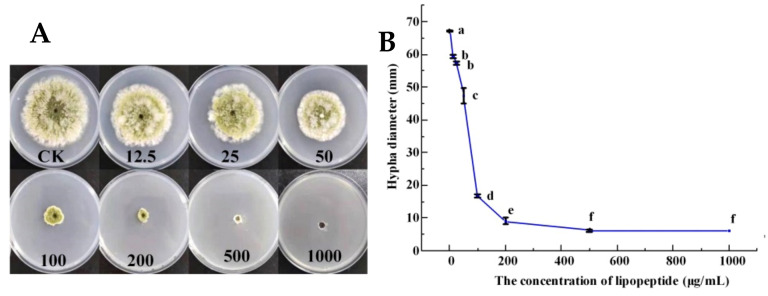
The changes in growth diameter and inhibition rate of *A. flavus* under the effect of the control group (SDW) and test group (12.5–1000 µg/mL, LPs) were treated, respectively. (**A**) Colony morphology of *A**. flavus* grown on media containing different concentrations of LPs. (**B**) Colony diameter of *A**. flavus* after treating with LPs. Vertical bars represent standard errors of the mean. Different letters indicated significant differences (LSD test, *p* < 0.05).

**Figure 5 foods-11-00140-f005:**
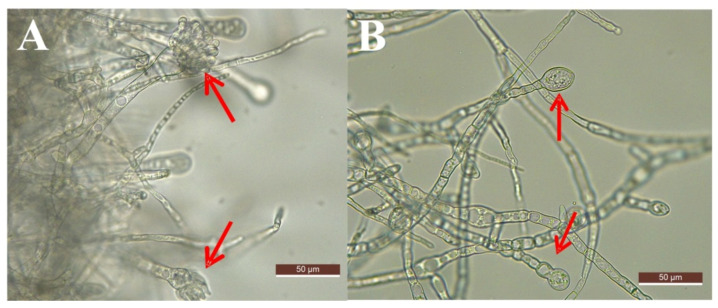
LPs released by Strain E2 exhibited an inhibitory effect on the growth of *A. flavus* (×400). (**A**): Control; (**B**): Treatment with 500 µg/mL LPs.

**Figure 6 foods-11-00140-f006:**
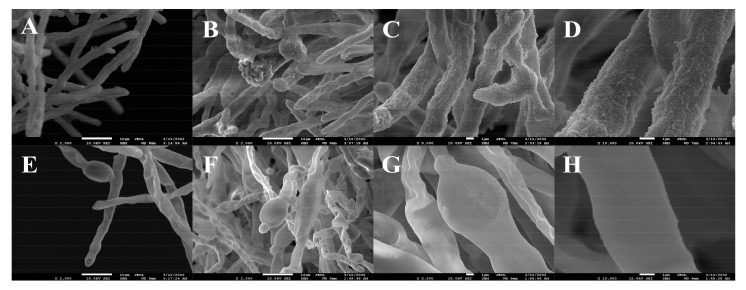
SEM of *A. flavus* growth without the presence of antagonistic substances (control: (**A**–**D**)) and faced with LPs (**E**–**H**). The figures show: suppression of mycelial growth of *A. flavus*, the apex of the mycelium was broken (**E**); degenerative change in the morphology of the fungus hyphae (**F**); the mycelium was twisted, collapsed, and abnormally swollen (**G**); the mycelia was smooth and more swollen than the control group (**H**).

**Figure 7 foods-11-00140-f007:**
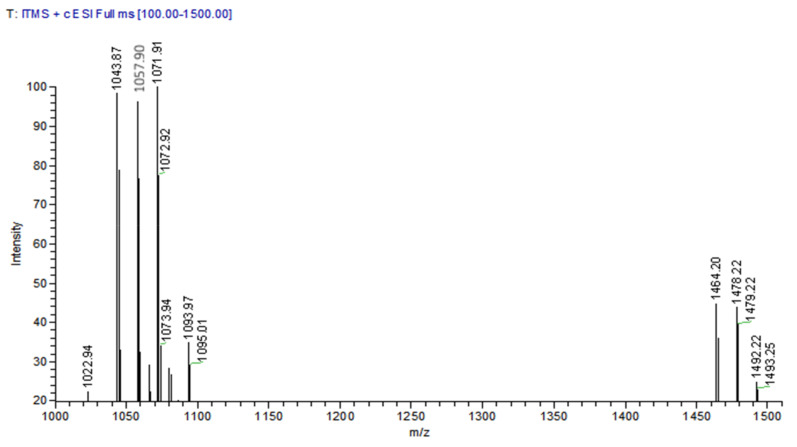
HPLC-MS for the LPs antibiotics isolated from strain E2.

**Figure 8 foods-11-00140-f008:**
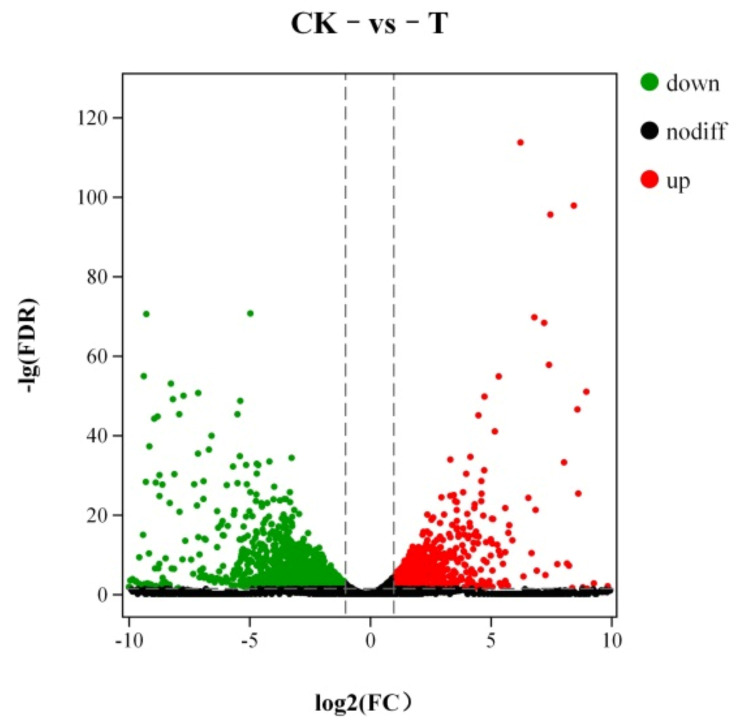
The volcano plot of differentially expressed genes.

**Figure 9 foods-11-00140-f009:**
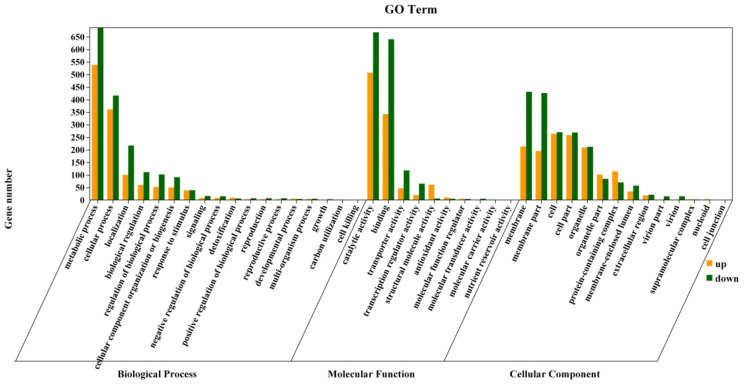
GO Analysis of differential genes of *A. flavus*.

**Figure 10 foods-11-00140-f010:**
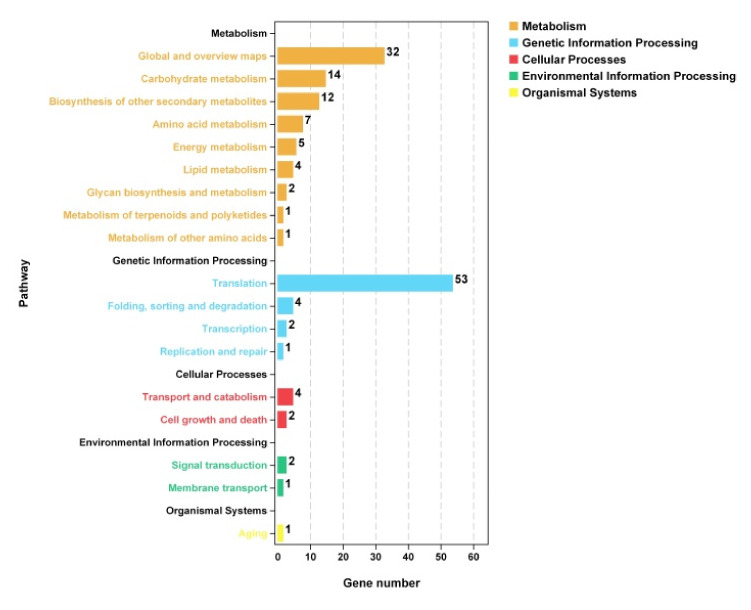
KEGG pathway classification of the screened differential genes.

**Figure 11 foods-11-00140-f011:**
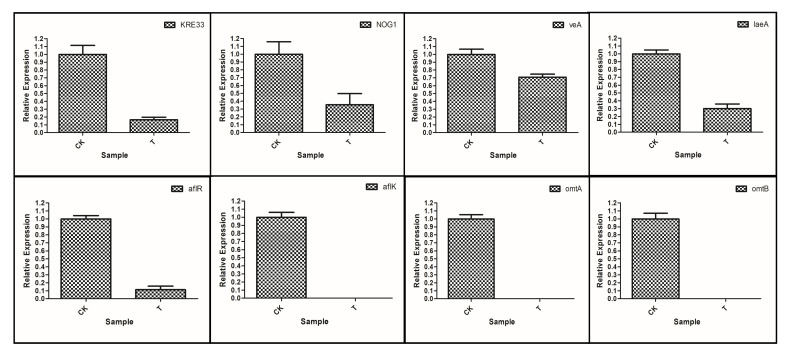
RT-qPCR verification results of some key differential genes. The actin gene was used as a reference gene. Vertical bars represent standard errors of the mean. (*p* < 0.05).

**Table 1 foods-11-00140-t001:** Primers used for RT-qPCR.

Gene ID	Symbol	Primers (5′ to 3′)
AFLA_029920	KRE33	F: GGGATTTACCAGTCACCGCA R: AGCACGATTCCACCTCCTTC
AFLA_134410	NOG1	F: TGCTCCAGAACCGTCAAACT R: CGGCTGAGAACGACATCCAA
AFLA_033290	laeA	F: CTTGCTCCCATACAGCCCTC R: TCCCATCACACTTCCACACC
AFLA_066460	veA	F: AAGGTTGTCGTGTGCGGATT R: TGGGGTAGAGATTCGGTCAG
AFLA_139190	aflK	F: GTGATTGAGGCGGGAGGAT R: GCCGTGTTGTCGTTGAGAG
AFLA_139360	aflR	F: CCCCACTACCACCGTTTCAG R: CTCATCCACACAATCCTCGC
AFLA_139210	omtA	F: TAGTTCATGGCCCGGTTCC R: AGGTTTGCCTTTCGTCTGCT
AFLA_139220	omtB	F: GAGAGCGACACGCCGATAA R: GAAGAATGCGACCAAGGAGT
AFLA_047410	actin	F: GAAGTTGCTGCTCTCGTCA R: GACCGACAATGGAGGGGAAG

**Table 2 foods-11-00140-t002:** Physiological and biochemical properties of strain E2.

Properties	Strain E2	Properties	Strain E2
Gram stain	+	Citrate utilization	+
Moveability	+	Glucose ferm entation	+
Catalase	+	Mannitol fermentation	+
V-P test	+	Carbon utilization–glucose	+
Methyl red test	−	Carbon utilization–mannitol	+
Starch hydrolusis	+	Carbon utilization–fructose	+
Gelaune liquefaction	+	Gelaune liquefaction	+

**Table 3 foods-11-00140-t003:** Effects of different LPs concentrations on spore germination.

Treatments	Lipopeptide (µg/mL)						
	CK	12.5	25	50	100	200	400
Spore germination rate (%)	75.34 ±1.07 a	67.41 ±1.82 b	54.93 ±1.33 c	35.49 ±3.14 d	15.43 ±2.62 e	5.80 ±1.31 f	1.33 ±0.52 f

Columns with different letters indicated significant differences (LSD test, *p* < 0.05).

**Table 4 foods-11-00140-t004:** LC-MS detection of purified LPs.

Retention Time (min)	MS *m/z*		Identified Compounds
	[M + H]+	[M + Na]+	
13.25	1043.87	1065.97	C14 Iturin A
15.99	1057.90	1079.96	C15 Iturin A
16.52	1057.90	1079.97	C15 Iturin A
20.95	1071.91	1093.97	C16 Iturin A
21.85	1071.92	1093.97	C16 Iturin A
24.89	1464.20	-	C16 FengycinA
27.20	1478.20	-	C16 FengycinC
30.61	1492.22	-	C16 FengycinD

**Table 5 foods-11-00140-t005:** Expression profiling of *A. flavus* genes involved in ribosome biogenesis in eukaryotes.

Gene ID	Gene Description	Log_2_FC	Style
AFLA_012380	U3 small nucleolar ribonucleoprotein protein Mpp10	−2.22	down
AFLA_029920	nucleolar ATPase Kre33, putative	−2.48	down
AFLA_033570	SSU processome component Utp10, putative	−2.22	down
AFLA_028940	small nucleolar ribonucleoprotein complex subunit Utp14	−2.88	down
AFLA_112310	small nucleolar ribonucleoprotein complex subunit, putative	−2.03	down
AFLA_113720	ribosome biogenesis (Nop4), putative	−2.30	down
AFLA_134410	nucleolar GTP-binding protein (Nog1), putative	−2.18	down
AFLA_110550	nucleolar GTPase, putative	−2.68	down

**Table 6 foods-11-00140-t006:** Expression profiling of *A. flavus* genes involved in aflatoxin biosynthesis.

Gene ID	Gene Description	Log_2_FC	Style
AFLA_138050	flavonoid 3-hydroxylase, putative	−5.60	down
AFLA_139140	aflYa/nadA/NADH oxidase	−4.45	down
AFLA_139190	aflK/vbs/VERB synthase	−9.25	down
AFLA_139200	aflQ/ordA/ord-1/oxidoreductase/cytochrome P450 monooxigenase	−9.27	down
AFLA_139210	aflP/omtA/omt-1/O-methyltransferase A	−8.93	down
AFLA_139220	aflO/omtB/dmtA/O-methyltransferase B	−8.79	down
AFLA_139260	aflG/avnA/ord-1/cytochrome P450 monooxygenase	−9.13	down
AFLA_139280	aflN/verA/monooxygenase	−9.14	down
AFLA_139300	aflM/ver-1/dehydrogenase/ketoreductase	−7.10	down
AFLA_139310	aflE/norA/aad/adh-2/NOR reductase/dehydrogenase	−8.23	down
AFLA_139320	aflJ/estA/esterase	−8.15	down
AFLA_139330	aflH/adhA/short chain alcohol dehydrogenase	−3.34	down
AFLA_139340	aflS/pathway regulator	−2.77	down
AFLA_139360	aflR/apa-2/afl-2/transcription activator	−2.12	down
AFLA_139370	aflB/fas-1/fatty acid synthase beta subunit	−2.55	down
AFLA_139390	aflD/nor-1/reductase	−4.68	down
AFLA_139410	aflC/pksA/pksL1/polyketide synthase	−3.13	down
AFLA_139420	aflT/aflT/transmembrane protein	−5.48	down
AFLA_066460	developmental regulator AflYf/VeA	−0.40	down
AFLA_033290	regulator of secondary metabolism LaeA	−1.63	down

## Data Availability

The data showed in this study are contained within the article.

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
