# Peer review of "Screening of Bacillus velezensis E2 and the Inhibitory Effect of Its Antifungal Substances on Aspergillus flavus"

_foods, 2022, doi:10.3390/foods11020140_

Round 1

Reviewer 1 Report

The authors isolated a new Bacillus velezensis strain (named E2) with significant biocontrol properties against Aspergillus flavus, the causal agent of mold in several post-harvest crops. The highlighted bioactivity seems to be due to lipopeptides (LPs) belonging to iturin and fengycin families, produced by the bacterium. Based on cytological and transcriptomic assays, the modes of action of the LPs on the pathogen seems to rely on the destruction of cell membrane integrity and interference with the normal functioning of ribosomes. Moreover, LPs likely down-regulate in A. flavus the expression of genes involved in aflatoxin (mycotoxin) biosynthesis, thereby making the identified bacterium as a promising agent for sustainable management of A. flavus.

Overall, the study is well conducted and the experiments sems to be well performed. The obtained results are for valuable importance and provide new insights into the biocontrol of A. flavus. However, below are some comments that should be considered by the authors to improve the quality of their manuscript :

Major comment : The authors detected 3164 differentially expressed genes (DEGs) under the criteria of Log2 (Fold Change)|≥1 and FDR<0.05. The threshold of ≥1 is too low for such a non-targeted transcriptomic analysis. The authors should apply a threshold of at least ≥2 with FDR<0.05. This could lead to a decrease of the number of highlighted DEGs, hence potentially impacting the conclusions.

- The writing quality of the manuscript could be improved by an additional academic reader and/or native English speaker for language quality. Some sentences should be reformulated to improve their clarity. Few minor suggestions (among others) are listed in the bottom.

- Materials and Methods section : Move the sub-section “2.3. Identification of strain E2 based on its 16S rDNA gene sequence” to 2.2. (before antifungal bioassays).

- How was the concentration of 1 mg/mL chosen for the transcriptomic assay? May be corresponds to a sub-lethal concentration? The authors need to justify this concentration in the manuscript.

- Figure 2 : Change N2 by E2 in the phylogenetic tree.

- One weakness of the study is that the authors tested only a mixture of LPs and they provide no data about the specific LP responsible for the observed antifungal activity. They at least should develop more the discussion part dedicated to this aspect (Lines 367 - 375).

Specific suggestions :

Lines 8-9 : Change the sentence to “Improving the grain drying process after harvest is essential to enhance the quality of rice production and storage.”

Line 10 : Delete “(A. flavus)”.

Line 10 : Change “mildew” by “mold”. Mildew is a disease occurring in field conditions.

Line 11 : “Bacillus sp.” in italic, here and throughout the whole manuscript.

Line 18 : Delete “in the transcription”

Line 29 : Change “A. flavus” by “Aspergilus flavus”

Line 49 : Change “Common” by “common”

Line 51 : Change “bacillus” by “Bacillus”

Lines 55-56 : Change the sentence to “Later, the antifungal substances (LPs) occurring in the fermentation broth were extracted”.

Lines 56-57 : Delete the sentence to “In the present work, we showed that 56 the effects of LPs minimum inhibitory concentration (MIC).”

Line 60 : Give full description of “SEM”

Line 65 : “A. flavus” in italic.

Line 178 : one reference gene “actin” is not enough for the reliability of the quantification. At least two housekeeping genes are generally recommended for qPCR assays.

Line 215 : Change LPS to LPs.

Line 192 : Change the sentence “…were screened on the rice surface, soil, and 192 paddy leaf surface.” To “…were isolated the rice surface, soil, and 192 paddy leaf surface.”.

Line 223 : You mean 75.34 as indicated in the Table 3 (and not 98.67 %) ?

Figures 5 and 6 : The scale bars are not visible.

Table 5 : Keep two numbers only after the comma in the “Log2FC” column.

Author Response

Point 1: The authors isolated a new Bacillus velezensis strain (named E2) with significant biocontrol properties against Aspergillus flavus, the causal agent of mold in several post-harvest crops. The highlighted bioactivity seems to be due to lipopeptides (LPs) belonging to iturin and fengycin families, produced by the bacterium. Based on cytological and transcriptomic assays, the modes of action of the LPs on the pathogen seems to rely on the destruction of cell membrane integrity and interference with the normal functioning of ribosomes. Moreover, LPs likely down-regulate in A. flavus the expression of genes involved in aflatoxin (mycotoxin) biosynthesis, thereby making the identified bacterium as a promising agent for sustainable management of A. flavus.

Overall, the study is well conducted and the experiments sems to be well performed. The obtained results are for valuable importance and provide new insights into the biocontrol of A. flavus.

Response 1: The authors greatly appreciate the positive general comments.

Point 2: The authors detected 3164 differentially expressed genes (DEGs) under the criteria of Log2 (Fold Change)|≥1 and FDR<0.05. The threshold of ≥1 is too low for such a non-targeted transcriptomic analysis. The authors should apply a threshold of at least ≥2 with FDR<0.05. This could lead to a decrease of the number of highlighted DEGs, hence potentially impacting the conclusions.

Response 2: We have revised the criteria of Log2 (Fold Change)|≥1 to Log2 (Fold Change).As the reviewer's comment, the differentially expressed genes we detected reduced to 1500 under this criteria. But, the most diverse genes in the biological process were still found in the metabolic process and cellular process. The most diverse genes in molecular function were still catalytic activity and binding. The only difference is that the number of differential genes has slightly reduced. Besides, other representative genes related to ribosome synthesis in eukaryotes, such as mpp10, SPAC57A7.06, KRE33, utp10, nog-2, utp13, and SPBC4F6.14, were still down-regulated. This result still helped us prove that LPs inhibited the synthesis of certain proteins and damaged Aspergillus flavus by down-regulating key genes related to ribosome synthesis in eukaryotes. We have revised the the relevant data(Figure 8, Figure 9, Figure 10, Table 5, Table 6, Line169-170, 287-289, 294-296, 303, 315-316).

Point 3: The writing quality of the manuscript could be improved by an additional academic reader and/or native English speaker for language quality. Some sentences should be reformulated to improve their clarity.

Response 3: The writing quality of the manuscript have been improved by a native English speaker for language quality. Some sentences have been reformulated to improve their clarity.

Point 4: Materials and Methods section : Move the sub-section “2.3. Identification of strain E2 based on its 16S rDNA gene sequence” to 2.2. (before antifungal bioassays).

Response 4: We need to screen out the antagonistic bacteria before we can be sure that the strain we want to identify is strain E2. We have merged 2.3 to 2.2

Point 5: How was the concentration of 1 mg/mL chosen for the transcriptomic assay? May be corresponds to a sub-lethal concentration? The authors need to justify this concentration in the manuscript.

Response 5: The LPs concentration of 1mg/mL was close to the sub-lethal concentration. Through the results of 3.2, it could be concluded that the inhibitory effect of 1mg/ml LPs on Aspergillus flavus is slightly inferior to that of the bacteria itself. However, under the action of lipopeptide at a concentration of 1mg/ml, the growth rate of Aspergillus flavus, the amount of mycelium, and the morphology of mycelium have changed significantly. Therefore, the concentration of 1mg/ml was used for the transcriptomic assay.

Point 6: Figure 2 : Change N2 by E2 in the phylogenetic tree

Response 6: We have changed N2 by E2 in the phylogenetic tree.

Point 7: One weakness of the study is that the authors tested only a mixture of LPs and they provide no data about the specific LP responsible for the observed antifungal activity. They at least should develop more the discussion part dedicated to this aspect (Lines 367 - 375).

Response 7: We have developed more the discussion part dedicated to this aspect (Lines 378 - 381, Lines 387 - 390).

Point 8: Lines 8-9 : Change the sentence to “Improving the grain drying process after harvest is essential to enhance the quality of rice production and storage.” 

Response 8: We have deleted the sentence.

Point 9: Line 10 : Delete “(A. flavus)”.

Response 9: We have deleted “(A. flavus)”.

Point 10: Line 10 :Change “mildew” by “mold”. Mildew is a disease occurring in field conditions.

Response 10: We have changed “mildew” by “mold”.

Point 11: Line 11 :“Bacillus sp.” in italic, here and throughout the whole manuscript.

Response 11: We have changed “Bacillus sp.” in italic, here and throughout the whole manuscript.

Point 12: Line 18 : Delete “in the transcription” .

Response 12: We have deleted “in the transcription”.

Point 13: Line 29 : Change “A. flavus” by “Aspergilus flavus”.

Response 13: We have changed “A. flavus” by “Aspergilus flavus”.

Point 14: Line 49 : Change “Common” by “common”.

Response 14: We have changed “Common” by “common”.

Point 15: Line 49 : Line 51 : Change “bacillus” by “Bacillus”.

Response 15: We have changed “bacillus” by “Bacillus”.

Point 16: Line 49 : Change the sentence to “Later, the antifungal substances (LPs) occurring in the fermentation broth were extracted”.

Response 16: We have changed the sentence to “Later, the antifungal substances (LPs) occurring in the fermentation broth were extracted”.

Point 17: Delete the sentence to “In the present work, we showed that 56 the effects of LPs minimum inhibitory concentration (MIC).”

Response 17: We have deleted the sentence to “In the present work, we showed that 56 the effects of LPs minimum inhibitory concentration (MIC).”

Point 18: Line 60 : Give full description of “SEM”.

Response 18: We have given full description of “SEM”(Lines 133-142).

Point 19: Line 49 :  A. flavus” in italic.

Response 19: We have changed “A. flavus” in italic.

Point 20: Line 49 : one reference gene “actin” is not enough for the reliability of the quantification. At least two housekeeping genes are generally recommended for qPCR assays.

Response 20: We selected one reference gene "actin" by referring to a large number of reports. We greatly appreciate this important comment. We realized that one reference gene was really not enough for the reliability of the quantification. Due to the revision deadline, we first submitted the revised draft, the trial is continuing, and will be added later.

Point 21: Line 215 : Change LPS to LPs.

Response 21: We have changed LPS to LPs.

Point 22: Line 192 : Change the sentence “…were screened on the rice surface, soil, and 192 paddy leaf surface.” To “…were isolated the rice surface, soil, and 192 paddy leaf surface.”.

Response 22: We have changed the sentence “…were screened on the rice surface, soil, and 192 paddy leaf surface.” To “…were isolated the rice surface, soil, and 192 paddy leaf surface.”

Point 23: Line 223 : You mean 75.34 as indicated in the Table 3 (and not 98.67 %) ?

Response 23: This is a mistake in our expression. The spore germination rate of the control group was 75.34%. We have changed “98.67 %” in “75.34%”.

Point 24: Line 49 : Figures 5 and 6 : The scale bars are not visible.

Response 24: The scale bars are at the bottom right of the figures, and we have increased the resolution of them.

Point 25: Line 49 : Table 5 : Keep two numbers only after the comma in the “Log2FC” column.

Response 25: We have kept two numbers only after the comma in the “Log2FC” column.

Reviewer 2 Report

The authors studied new isolates of the genus Bacillus that show an inhibitory effect on  Aspergillus flavus. The most efficient isolate has been assigned by 16rDNA taxonomy to the species B. velenzensis, which is not a surprise considering the well describe antifungal strain B. velezensis FZB24. However, the data on the inhibitory impact on A. flavus is solid and of interest for the public and the scientific community.
I am not a native speaker, but to my opinion thew article is written in a poor english and should be completely rewritten with the help of a native speaker.

Major concerns:

The analysis of the isolated strain E2 is incomplete. In the manuscript the authors speculate that E2 produces two LPs because of the observed number of compounds within their experiment. The study would strongly benefit if the authors add the genome sequence of E2. The complete genome it enable the author to switch from speculation to solid data that would indicate the real potential of their isolate.

Minor concerns:

The authors mixed up taxonomic terms for instance lines Lines 39 to 52:
"Bacillus" is not a general term but a genus. The usage of strain which is a sub- entity of species by the authors is misleeding. Line 46: Antifugal LPs are not produced by all strains of all isolates from the genus Bacillus. So it should be written more precisely. For instance "produce by some strains of the genus Bacillus"
Line 50 relace "Bacillus" by "Some isolates"

line 160: The authors describe that they determined the integrity of the RNA by a Nanodrop. Nanodrop devices determine conteration and purity of RNA but are incapable of determining reilable integrity measures like RIN values.

Author Response

Point 1: The authors studied new isolates of the genus Bacillus that show an inhibitory effect on  Aspergillus flavus. The most efficient isolate has been assigned by 16rDNA taxonomy to the species B. velenzensis, which is not a surprise considering the well describe antifungal strain B. velezensis FZB24. However, the data on the inhibitory impact on A. flavus is solid and of interest for the public and the scientific community.

I am not a native speaker, but to my opinion thew article is written in a poor english and should be completely rewritten with the help of a native speaker.

Response 1: Thank you for the general comments. The manuscript has been critically read and edited by a native English speaker for language quality. Some sentences have been reformulated to improve their clarity.

Point 2: The analysis of the isolated strain E2 is incomplete. In the manuscript the authors speculate that E2 produces two LPs because of the observed number of compounds within their experiment. The study would strongly benefit if the authors add the genome sequence of E2. The complete genome it enable the author to switch from speculation to solid data that would indicate the real potential of their isolate.

Response 2: We have added the genome sequence of E2 in the attachment

Point 3: The authors mixed up taxonomic terms for instance lines Lines 39 to 52:

"Bacillus" is not a general term but a genus. The usage of strain which is a sub- entity of species by the authors is misleeding. Line 46: Antifugal LPs are not produced by all strains of all isolates from the genus Bacillus. So it should be written more precisely. For instance "produce by some strains of the genus Bacillus" 

Response 3: We have revised taxonomic terms for instance lines Lines 39 to 52 and throughout the whole manuscript.

Point 4: Line 50 relace "Bacillus" by "Some isolates".

Response 4: We have changed "Bacillus" by "Some isolates".

Point 5: The authors describe that they determined the integrity of the RNA by a Nanodrop. Nanodrop devices determine conteration and purity of RNA but are incapable of determining reilable integrity measures like RIN values.

Response 5: We confirmed to Gene Denovo Biotechnology Co. (Guangzhou, China) that the transcriptome used Agilent 2100 to detect samples. Agilent 2100 are capable of determining reilable integrity measures like RIN values. We have changed "Nanodrop" to "Agilent 2100" "Line162".